# The Spatiotemporal Dynamics of Facial Movements Reveals the Left Side of a Posed Smile

**DOI:** 10.3390/biology12091160

**Published:** 2023-08-23

**Authors:** Elisa Straulino, Cristina Scarpazza, Andrea Spoto, Sonia Betti, Beatriz Chozas Barrientos, Luisa Sartori

**Affiliations:** 1Department of General Psychology, University of Padova, Via Venezia 8, 35131 Padova, Italy; cristina.scarpazza@unipd.it (C.S.); andrea.spoto@unipd.it (A.S.); 2Translational Neuroimaging and Cognitive Lab, IRCCS San Camillo Hospital, Via Alberoni 70, 30126 Venice, Italy; 3Department of Psychology, Centre for Studies and Research in Cognitive Neuroscience, University of Bologna, Viale Rasi e Spinelli 176, 47521 Cesena, Italy; sonia.betti2@unibo.it; 4Department of Chiropractic Medicine, University of Zurich, Balgrist University Hospital, Forchstrasse 340, 8008 Zürich, Switzerland; beatriz.chozasbarrientos@balgrist.ch; 5Padova Neuroscience Center, University of Padova, Via Giuseppe Orus 2, 35131 Padova, Italy

**Keywords:** emotion expressions, kinematics, lateralization, happiness, emotional induction, motor contagion, dynamic patterns

## Abstract

**Simple Summary:**

Humans have the amazing ability to make thousands of different facial expressions due to the existence of two different brain pathways for facial expressions: The Voluntary Pathway, which controls intentional expressions, and the Involuntary Pathway, which is activated for spontaneous expressions. These two pathways could also differentially influence the left and right sides of the face when we make a posed smile or a spontaneous smile, an issue that has not been studied carefully before. In two experiments, we found a double-peak pattern: compared to the felt smile, the posed smile involves a faster and wider movement in the left corner of the mouth, while an early deceleration of the right corner occurs in the second phase of the movement, after the speed peak. Our findings will aid to clarify the lateralized bases of emotion expression.

**Abstract:**

Humans can recombine thousands of different facial expressions. This variability is due to the ability to voluntarily or involuntarily modulate emotional expressions, which, in turn, depends on the existence of two anatomically separate pathways. The Voluntary (VP) and Involuntary (IP) pathways mediate the production of posed and spontaneous facial expressions, respectively, and might also affect the left and right sides of the face differently. This is a neglected aspect in the literature on emotion, where posed expressions instead of genuine expressions are often used as stimuli. Two experiments with different induction methods were specifically designed to investigate the unfolding of spontaneous and posed facial expressions of happiness along the facial vertical axis (left, right) with a high-definition 3-D optoelectronic system. The results showed that spontaneous expressions were distinguished from posed facial movements as revealed by reliable spatial and speed key kinematic patterns in both experiments. Moreover, VP activation produced a lateralization effect: compared with the felt smile, the posed smile involved an initial acceleration of the left corner of the mouth, while an early deceleration of the right corner occurred in the second phase of the movement, after the velocity peak.

## 1. Introduction

The human face has 43 muscles, which can stretch, lift, and contort into thousands combinations involving different muscles at different times and with different intensities [1]. In neuroanatomical terms, movement of the human face is controlled by two cranial nerves, the facial nerve (cranial nerve VII) and the trigeminal nerve (cranial nerve V). The facial nerve controls the superficial muscles attached to the skin, which are primarily responsible for facial expressions, and originate in the two facial nuclei located on either side of the midline in the pons. These nuclei do not communicate directly with each other, and this is why emotional expression can vary in intensity across a vertical axis (i.e., left vs. right). Differences can also occur across a horizontal axis (i.e., upper vs. lower area), given that the facial nerve has five major branches, with each branch serving a different portion of the face. In particular, the upper face (i.e., eye area) is controlled differently than the lower part (i.e., mouth area [2,3]). The upper part of the face receives input from both the ipsilateral and contralateral facial nerves, whereas the lower part is controlled primarily by the contralateral facial nerve [4,5]. Differences in the lateralization of facial expressions may therefore result from a lack of communication between the facial nuclei. These differences are mainly observed in the lower part of the face, due to the contralateral innervation produced by the branch of the facial nerve responsible for the contraction of the muscles around the mouth. These asymmetries should be specifically amplified during posed expressions of happiness, which are controlled by the Voluntary Pathway. Indeed, emotional expressions can be voluntarily or involuntarily modulated depending on the recruitment of two anatomically separate pathways for the production of facial expressions: the Voluntary Pathway (VP) and the Involuntary Pathway (IP [6]). The former involves input from the primary motor cortex and is primarily responsible for voluntary expression. The second, on the other hand, is a subcortical system that is primarily responsible for spontaneous expression. The contraction of facial muscles related to genuine emotion originates from subcortical brain areas that provide excitatory stimuli to the facial nerve nucleus via extrapyramidal motor tracts. In contrast, posed expressions are controlled by impulses of the pyramidal tracts from the primary motor cortex [7,8,9]. Therefore, small changes in the dynamical development of a facial display may characterize and distinguish genuine from posed facial expressions in each of the four quadrants resulting from the intersection of the vertical and horizonal axes, a topic so far neglected (but see [5,10]).

In this respect, three major models of emotional processing address the so-called “hemispheric lateralization of emotion” topic in humans [11,12]: the Right-Hemisphere Hypothesis [8], the Valence-Specific Hypothesis [13], and the Emotion-type Hypothesis [5,14]. Analysis of facial expressions has been a traditional means for inferring hemispheric lateralization of emotions by measuring expressive differences between the left and right hemiface, based on the assumption that the right hemisphere controls the left side of the face, and the left hemisphere controls the right side of the face [14,15,16,17,18,19,20]. The Right-Hemisphere Hypothesis [8] states that all emotions are a dominant, lateralized function of the right hemisphere, regardless of their valence or the emotional feeling processed. Their associated expressions would therefore be lateralized in the left side of the face. The Valence-Specific Hypothesis [13] states that negative, avoidance, or withdrawal-type emotions are lateralized to the right hemisphere (the associated expressions would then be lateralized to the left hemiface), whereas positive emotions, such as happiness, are lateralized to the left hemisphere (with expressions lateralized to the right hemiface). Finally, the Emotion-type Hypothesis [21] states that primary emotional responses are initiated by the right hemisphere on the left side of the face, while social-emotional responses are initiated by the left hemisphere on the right side of the face. Primary emotions and their manifestations are happiness, sadness, anger, fear, disgust, and surprise; whereas social emotions, such as embarrassment, envy, guilt, and shame, are acquired through parental socialization and during play, school, and religious-cultural activities [21].

Based on these theories, various patterns of facial lateralization of emotion expressions can be hypothesized (for a schematic representation of different hypotheses and related predictions on happiness expressions, see Figure 1a). However, a large number of replication studies exploring those hypotheses provided inconsistent results [11,12]. One possible explanation for these contradictory data is that previous literature considered both genuine and posed expressions without distinction. This aspect is particularly remarkable when considering expressions of happiness. Duchenne and non-Duchenne are terms used to classify if a smile reflects a true emotional feeling versus a false smile [22,23,24]. A felt (Duchenne) smile is very expressive and it is classically described as causing the cheeks to lift, the eyes to narrow, and wrinkling of the skin to produce crow’s feet. A false (non-Duchenne) smile, instead, would only involve the lower face area. However, recent research using high-speed videography has shown that the difference between a felt (Duchenne) versus a fake smile might in fact relate to the side of the face initiating the smile [25], thus providing a fundamental—yet not tested—cue to an observer for emotion authenticity detection. The key role of temporal features (i.e., time-onset vs maximum expression) as a locus for investigating the lateralization of facial displays and inferring hemispheric lateralization has been, in fact, largely neglected. A rigorous methodological approach able to track the full unfolding of an expression over time and across the two spatial axes is therefore necessary to characterize and distinguish spontaneous from posed smiles and to provide crucial cues for the hemispheric lateralization debate.

A major drawback of the existing literature on facial expressions is that, to collect reliable and controlled databases, researchers typically showed participants static images of posed expressions (for reviews, see [15,26]). Adopted stimuli were neither dynamic nor genuine, and the induction method [27] did not differentiate between Emotional Induction (i.e., the transmission of emotions from one individual to another [28,29]) and Motor Contagion (i.e., the automatic reproduction of the motor patterns of another individual [30,31]). To conclude, inconsistencies in the major models of hemispheric lateralization of emotion and arbitrary use of different experimental stimuli and elicitation methods are all sources of poor consensus in the literature on facial expressions of emotion.

The objective of this study was to investigate lateralized patterns of movement in the expression of happiness and the possible impact of dynamic stimuli with different induction methods on spontaneous expressions (i.e., Emotional Induction, Motor Contagion). In particular, we hypothesized that lateralized kinematic patterns should have emerged in the lower part of the face for posed expressions innervated contralaterally by the VP and that Motor Contagion should have modified the choreography of spontaneous expressions.

To investigate these hypotheses, in Experiment 1, we presented two sets of stimuli: (i) videoclips extracted from popular comedies that produced hilarity without showing smiling faces (Spontaneous condition, Emotional Induction), and (ii) static pictures of smiles (Posed condition). In Experiment 2, we showed videos of people shot frontally while manifesting the expression of happiness (Spontaneous condition, Motor Contagion). For the Posed condition, we maintained the same procedure as in Experiment 1.

To test the spatiotemporal dynamics of facial movements, we capitalized on a method recently developed in our laboratory [32], which combines an ultra-high definition optoelectronic system with a Facial Action Coding System (FACS, a comprehensive, anatomically based system for describing all visually discernible facial movement) [33]. This method proved to be remarkably accurate in the quantitative capture of facial motion.

Thanks to this method, in Experiments 1 and 2, we expected to show lateralized kinematic patterns in the lower part of the face for posed compared to spontaneous expressions. On the other hand, we expected to find differences for spontaneous expressions across the two experiments depending on the induction method.

Finally, we tried to disambiguate which hypothesis on the hemispheric lateralization of emotion expressions is more rigorous in explaining the observed data.

## 2. General Methods

The data for Experiments 1 and 2 were collected at the Department of General Psychology—University of Padova.

### 2.1. Ethics Statement

All experiments were conducted in accordance with the Declaration of Helsinki and approved by the Ethics Committee of the University of Padova (protocols no. 3580, 4539). All participants were naïve to the purposes of the experiment and gave their written informed consent for their participation.

### 2.2. Apparatus

Participants were tested individually in a dimly lit room. Their faces were recorded frontally with a video camera (Logitech C920 HD Pro Webcam, Full HD 1080p/30fps Logitech, Lausanne, CH) positioned above the monitor for the FACS validation procedure. The stimuli presentation was implemented using E-prime V2.0. Five infrared reflective markers (i.e., ultra-light 3 mm diameter semi-spheres) were applied to the faces of participants according to the Clepsydra Model (Figure 1b, top picture [34]) for kinematic analysis. We selected the minimum number of markers adopted in the literature as a common denominator to compare our findings with previous results [35,36]. Markers were taped to the left and right eyebrows and to the left and right cheilions to test the facial nerve branches that specifically innervate the upper and the lower parts of the face, respectively. A further marker was placed on the tip of the nose to perform a detailed analysis of the lateralized movements of each marker with respect to this reference point. The advantage of applying kinematic analysis to pairs of markers rather than individual markers is that it accounts for any head movement [37]. Because of its simplicity, the Clepsydra model could be validated and replicated by different laboratories around the world [38]. Six infrared cameras (sampling rate 140 Hz), placed in a semicircle at a distance of 1–1.2 m from the center of the room (Figure 1b, bottom picture) captured the relative position of the markers. Facial movements were recorded using a 3-D motion analysis system (SMART-D, Bioengineering Technology and Systems [B|T|S] BTS, Milano, Italy). The coordinates of the markers were reconstructed with an accuracy of 0.2 mm over the field of view. The standard deviation of the reconstruction error was 0.2 mm for the vertical (Y) axis and 0.3 mm for the two horizontal (X and Z) axes.

### 2.3. Procedure

Each participant underwent a single experimental session (Experiment 1 or 2) lasting approximately 20 min. They were seated in a height-adjustable chair in front of a monitor (40 cm from the edge of the table) and were free to move while observing selected stimuli displayed on the monitor (Figure 2). Facial movements were recorded under two experimental conditions: (i) Spontaneous condition, in which participants watched happiness-inducing videos and reacted freely (i.e., they were given no instructions); (ii) Posed condition, in which participants produced a voluntary expression of happiness while a posed image of happiness was shown on the monitor. The two experimental conditions within each experiment—Spontaneous and Posed—were specifically adopted to activate the two pathways (Voluntary and Involuntary). Crucially, we wanted to test the two methods of spontaneous induction in two separate experiments to avoid possible carry-over effects between them while comparing them both with the same Posed condition. This condition served, on the one hand, to define each participant’s expressive baseline (as a term of intra-individual comparison) and, on the other hand, to test the specific role of the Voluntary versus Involuntary Pathway. Moreover, for the Posed condition, we chose a classical image of happiness taken from Ekman’s dataset for three reasons: (i) for comparison with previous literature [39]; (ii) for relevance to our experimental manipulation (being a prototypical non-genuine expression of happiness); and (iii) to keep the attention fixed on the monitor. Participants were instructed to mime the happiness expression three times so that we could have a sufficient number of repetitions. No instruction whatsoever was given on the duration of the expression. This procedure was aimed at generating expressions without forcing the participants to respect time constraints as in the Spontaneous conditions [40]. To avoid possible carry-over effects between trials due to the Emotional Induction from the videos used in the Spontaneous condition, we capitalized on the procedure adopted by Sowden and colleagues [35], and divided the trials into two separate blocks (first the spontaneous block, then the posed block after a brief pause). The inter stimulus interval was between 30 and 60 s.

### 2.4. Expression Extraction and FACS Validation Procedure

All repeated expressions of happiness within a single trial were included in the analysis. A two-step procedure was adopted to ensure a correct selection of each expression. First, we manually identified all the single epochs—the beginning and end of each smile—according to the FACS criteria (e.g., Action Units 6 and 12, the Cheek Raiser and the Lip Corner Puller). Despite the existence of an automated FACS coding, we decided to apply the manual one, as it is demonstrated to have a strong concurrent validity with the automated FACS coding, thus denoting the reliability of the method. Furthermore, the manual procedure has recently been demonstrated to outperform the automated one [41]. Second, we applied a kinematic algorithm to automatically identify the beginning and end of each smile using the cross-reference on the threshold velocity of the cheilion. Identification of motion onset and end performed with the two methods were compared and obtained a 100% match.

### 2.5. Data Acquisition

#### 2.5.1. Kinematic 3-D Tracking

Following kinematic data collection, the SMART-D Tracker software package (Bioengineering Technology and Systems, B|T|S) was employed to automatically reconstruct the 3-D marker positions as a function of time. Then, each clip was individually checked for correct marker identification.

#### 2.5.2. Kinematic 3-D Analysis

To investigate spatial, velocity, and temporal key kinematic parameters in both the upper and lower face, we considered the relative movement of two pairs of markers:
Lower part of the face:Left cheilion and the tip of the nose (Left-CH);Right cheilion and the tip of the nose (Right-CH).Upper part of the face:Left eyebrow and the tip of the nose (Left-EB);Right eyebrow and the tip of the nose (Right-EB).

Each expression was analyzed from the onset point to the apex (i.e., the peak). Movement onset was calculated as the first time point at which the mouth widening speed crossed a 0.2 mm/s threshold and remained above it for longer than 100 ms. Movement end was considered when the lip corners reached the maximum distance (i.e., the time at which the mouth widening speed dropped below the 0.2 mm/s threshold). Movement time was calculated as the temporal interval between movement onset and movement offset. We measured morphological (i.e., spatial) and dynamic (i.e., velocity and temporal) characteristics of each expression on each pair of markers [42]:Spatial parameters:Maximum Distance (MD, mm) is the maximum distance reached by the 3-D coordinates (x,y,z) of two markers.Delta Distance (DD, mm) is the difference between the maximum and the minimum distance reached by two markers, to account for functional and anatomical differences across participants.Velocity parameters:Maximum Velocity (MV, mm/s) is the maximum velocity reached by the 3-D coordinates (x,y,z) of each pair of markers. In the equation V = d/t, V is the velocity, d is the distance, and t is the time. The velocity of a pair of markers is calculated instant by instant as the displacement between the markers divided by the time required to make the displacement. The maximum velocity was the highest value of this equation and reflected the speed at which the two markers achieved maximum displacement in the minimum time (see Figure 3, blue line).Maximum Acceleration (MA, mm/s^2^) is the maximum acceleration reached by the 3-D coordinates (x,y,z) of each pair of markers. In the equation A = v/t, A is the acceleration, v is the velocity, and t is the time. Acceleration is calculated moment by moment as the rate of change of velocity of a pair of markers. The maximum acceleration was the highest value of this equation (see Figure 3, red dashed line).Maximum Deceleration (MDec, mm/s^2^): is the maximum deceleration reached by the 3-D coordinates (x,y,z) of each pair of markers. Deceleration is a negative acceleration and is calculated moment by moment as the rate of change of velocity of a pair of markers as their speed decreases. The maximum deceleration was the highest negative value of this equation, reported here in absolute value for graphical purposes (see Figure 3, red dashed line).

The time parameters were calculated by measuring the time when the spatial and velocity parameters just described reached their peaks after movement onset. Each temporal value was then normalized (i.e., divided by its corresponding total movement time) to account for individual speed differences:Time to Maximum Distance (TMD%, the proportion of time at which a pair of markers reached a maximum distance, calculated from movement onset)Time to Maximum Velocity (TMV%, the proportion of time at which a pair of markers reached a peak velocity, calculated from movement onset)Time to Maximum Acceleration (TMA%, the proportion of time at which a pair of markers reached a peak acceleration, calculated from movement onset)Time to Maximum Deceleration (TMDec%, the proportion of time at which a pair of markers reached a peak deceleration, calculated from movement onset)

### 2.6. Statistical Approach

All behavioral data were analyzed using JASP version 0.16 [44] statistical software. Data analysis for each experiment was divided into three main parts: the first one was aimed at testing whether facial motion differs across the vertical axis (i.e., Left-CH vs. Right-CH and Left-EB vs. Right-EB) for spontaneous and posed emotional expressions; the second part was aimed at exploring differences in the induction methods. During the first part of the analysis, for each experiment, a repeated-measures ANOVA with condition (Spontaneous, Posed) and side of the face (left, right) as within-subject variables was performed together with planned orthogonal contrasts. The Volk–Selke Maximum p-Ratio on the two-sided p-value was computed, too, in order to quantify the maximum possible odds in favor of the alternative hypothesis over the null one (VS-MPR [45]). Finally, to explore the possible differences triggered by different induction methods in the expression of happiness (posed and spontaneous), we conducted a mixed analysis of variance with Experiment (1, 2) as the between-subjects factor, and condition (Spontaneous, Posed), and side of the face (left, right) as the within-subjects factor. For all statistical analyses, a significance threshold of *p* < 0.05 was set and Bonferroni correction was applied to post hoc contrasts.

Sample size was determined by means of GPOWER 3.1 [46] based on previous literature [47]. Since we used a repeated-measures design in Experiments 1 and 2, we considered an effect size f of 0.25, alpha = 0.05, and power = 0.8. The projected sample size needed with this effect size was *N* = 20 for within-group comparisons in each experiment. For the comparison analysis, the sample obtained was the sum of the samples from Experiments 1 and 2, and, for this reason, it was not estimated a priori. We then calculated the post hoc power and found that, even with small effects, the power was high, namely > 0.95.

## 3. Experiment 1

### 3.1. Participants

Twenty participants were recruited to take part to the experiment. Three participants were subsequently excluded due to technical or recording problems; therefore, a sample of seventeen participants (13 females, 4 males) aged between 21 and 32 years (*M* = 24.8, *SD* = 3) were included in the analysis.

### 3.2. Stimuli

For the Spontaneous condition, we selected *N* = 2 emotion-inducing videos from a recently-validated dataset structured to elicit genuine facial expressions [40]. Videoclips were extracted from popular comedy movies in which actors produced hilarity without showing smiling faces (e.g., jokes by professional comedians). Videoclips lasted an average of 2 min and 55 s (video 1 = 3 min 49 s; video 2 = 2 min 2 s). The length of the clips did not exceed 5 min according to the recommended size for emotional video [48]. Each video was presented once without repetition to avoid possible habituation effects. Participants rated the intensity of the emotion felt while watching the videos at the end of each presentation. Participants rated the stimuli on a 9-point Likert scale, where 1 was negative, 5 was neutral, and 9 was positive. The mean score assigned to the stimuli (6; *SD* = 1.286) was significantly higher than the central value of the Likert scale (i.e., 5; t_17_ = 2.383; *p* = 0.015).

### 3.3. Results

Participants performed a range of 3–5 expressions of happiness per trial in the Spontaneous condition and three in the Posed condition.

#### Repeated-Measures ANOVA

In the lower part of the face, all the spatial and velocity kinematic parameters, together with two of temporal parameters (TMD%, TMA%), showed a main effect of condition (Posed vs. Spontaneous). In the upper part of the face, MD and TMV% showed a main effect of condition (Table 1). In general, the results showed an amplified choreography for posed expressions in spatial, velocity, and temporal terms compared to spontaneous expressions: posed smiles were wider, quicker, and more anticipated than spontaneous smiles (see graphical representation in Appendix A). A main effect of side of the face (Left vs. Right) was found on the lower part of the face for TMV% and TMDec% (Figure 4). In particular, the left cheilion reached its peak Velocity earlier than the right cheilion, and the right cheilion reached its Maximum Deceleration earlier than the left cheilion in both conditions.

## 4. Experiment 2

In this experiment we specifically manipulated the induction method to evaluate the effect of Motor Contagion (i.e., the automatic reproduction of the motor patterns of another individual) on the spontaneous expressions of happiness. While, in Experiment 1, happiness was induced with movie scenes showing professional actors who performed hilarious scenes without exhibiting smiling faces, in Experiment 2, we selected videos from YouTube in which people were shot frontally while being particularly happy and expressing uncontrollable laughter.

### 4.1. Participants

Twenty participants (15 females, 5 males) aged between 21 and 27 years (*M* = 23, *SD* = 1.8) were recruited to take part to the experiment.

### 4.2. Stimuli

The image adopted for the Posed condition was the same as for Experiment 1. Spontaneous happiness was instead elicited by using three emotion-inducing videos extracted from YouTube and already validated in a previous study from our laboratory [32]. While Experiment 1 videos were longer because the actor needed time to deliver the hilarious joke, in Experiment 2, the videos were shorter because only the expression of happiness was presented. As a result, the time available for participants to spontaneously smile while watching the videos was shorter. We therefore increased the number of stimuli from two to three to collect enough observations. Videoclips lasted an average of 49 s (video 1 = 31 s; video 2 = 57 s; video 3 = 59 s). Each video was presented once without repetition to avoid possible habituation effects. As in Experiment 1, participants rated the intensity of the emotion felt while watching the videos at the end of each presentation stimuli on a 9-point Likert scale, where 1 was negative, 5 was neutral, and 9 was positive. The mean score assigned to the stimuli (7; SD = 1.457) was significantly higher than the central value of the Likert scale (i.e., 5; t_19_ = 5.679; *p* < 0.001) and higher than the score reported in Experiment 1 (t_36_ = −2.535; *p* = 0.008).

### 4.3. Results

Participants performed a range of 3–5 expressions of happiness per trial in the Spontaneous condition and three in the Posed condition.

#### Repeated-Measures ANOVA

In the lower part of the face, all the kinematic parameters showed a main effect of condition (Posed vs. Spontaneous), except for TMV%. In the upper part of the face, no parameters showed any statistically significant effect (all *p_s_* > 0.05; Table 2). In general, the results confirmed the amplified choreography for posed expressions found in Experiment 1 for spatial, velocity, and temporal parameters compared to spontaneous expressions (for a graphical representation of the main effects of condition, see Appendix A). A main effect of side of the face (Left vs. Right) was shown for TMD%, TMA%, and TMDec%, and a statistically significant interaction condition by side of the face was found for MD and TMDec% (Figure 5 and Table 2). The results of the interaction showed that the left cheilion during posed expressions was more distal than the right cheilion during spontaneous expressions (Figure 5a). Crucially, the peak Deceleration of the right cheilion during posed expressions occurred earlier than during spontaneous expressions, and earlier than the peak of the left cheilion during posed smiles (Figure 5d). The results of the main effects showed that the left cheilion reached its Maximum Acceleration earlier than the right cheilion in both conditions (Figure 5c), but it reached its Maximum Distance later than the right cheilion in both conditions (Figure 5b).

## 5. Comparison Analysis (Experiment 1 vs. 2)

### Mixed ANOVA: Posed vs. Spontaneous, Left vs. Right, and Experiment 1 vs. 2

When directly comparing the possible differences triggered by different induction methods in the expression of happiness (posed and spontaneous), the variable Experiment (1 vs. 2) was never found to be significant, and, consequently, neither was the 3-way interaction between condition, side of the face and experiment for all investigated variables. However, a statistically significant main effect of condition (Posed vs. Spontaneous) was found in the lower part of the face for all the kinematic parameters, except for TMV%. In the upper part of the face, only MD showed a main effect of condition (see Table 3). In general, considering both Experiments 1 and 2, the results confirmed the amplified choreography for posed expressions for spatial, velocity, and temporal parameters compared to spontaneous expressions (for a graphical representation of the main effects of condition, see Appendix A). A main effect of side of the face (Left vs. Right) was shown for MD, TMV%, and TMDec%, and a statistically significant interaction condition by side of the face was found for MD and TMDec% (Figure 6 and Table 3). The results of the interaction showed that, during posed expressions, the left cheilion was more distal than the right cheilion (Figure 6a). Crucially, the peak Deceleration of the right cheilion during posed expressions occurred earlier than during spontaneous expressions, and earlier than the peak of the left cheilion during posed smiles (Figure 6c). Moreover, during spontaneous expressions, the peak Deceleration of the right cheilion occurred earlier than the peak of the left cheilion (Figure 6c). The results of the main effects showed that the left cheilion reached its Maximum Acceleration earlier than the right cheilion in both conditions (Figure 6c), but it reached its Maximum Distance later than the right cheilion in both conditions (Figure 6b).

## 6. Discussion

Facial expressions are a mosaic phenomenon, in which there is independent motor control of upper and lower facial expressions and a partially independent hemispheric motor control of the right and left sides [5]. Here, with two experiments, we reliably confirmed that facial movements provide relevant and consistent details to characterize and distinguish between spontaneous and posed expressions. In particular, the comparison analysis showed that a posed expression of happiness is characterized by increased peak distance of both cheilions and eyebrows, and increased peak velocity, acceleration, and deceleration of the cheilions compared to a spontaneous expression. In temporal terms, posed smiles show anticipated acceleration and deceleration peaks and a delayed peak distance compared to spontaneous smiles. These results were extended by showing also a lateralization pattern in spatiotemporal terms for posed expressions. The peak Distance, the Time to peak Velocity, and the Time to peak Deceleration appear, in fact, to be reliable markers of differences across the facial vertical axis. The peak Distance was increased, and the Velocity peak was reached earlier in the left side of the mouth compared to the right side. Whereas, in the second phase of the movement, after the velocity peak, an early Deceleration occurred in the right corner of the mouth. These data seem to indicate that the complex choreography of a fake smile implies a spatial amplification of the movement in the left hemiface, which then cascades into a slowdown in the final phase. More importantly, they tell us that this effect had a double peak across the hemiface, with the right corner of the mouth reaching earlier the peak Deceleration. In the case of a spontaneous smile, we find a lateralized effect only for one temporal component: the right corner of the mouth reached the Deceleration peak earlier with respect to the left side of the mouth.

### 6.1. Left vs. Right

In 2016, Ross and colleagues [5] were the first to describe a double-peak phenomenon. They reported that, in some cases, emotion-related expressions showed a slight relaxation before continuing to the final peak, and, in about one third of these expressions, the second innervation started on the contralateral side of the face. These qualitative observations led the authors to believe that the expressions were the result of two innervations, a “double-peak vertical blend”, indicating that the expression of interest had two independent motor components driven by opposite hemispheres. In the present study, in line with the hypothesis by Ross and colleagues [21], we found a consistent lateralization pattern in the left lower hemiface specific to posed expressions of happiness. That is, an acceleration that began first in the left corner of the mouth until the peak of maximum speed, beyond which an anticipated peak of Deceleration occurred in the right corner of the mouth.

The adoption of a novel 3-D kinematic approach allowed us to investigate the morphological and dynamic characteristics of lateralized expressions on each of the four quadrants resulting from the vertical and horizontal axes. Notably, previous studies using 2-D automated facial image analysis (e.g., [49]) found no evidence of asymmetry between the left and right side of a smile, likely due to a methodological limitation. Accurate assessment of asymmetry requires, in fact, either a frontal view of the face or precise 3-D registration. Moreover, a 3-D dynamic analysis is also required to exclude asymmetries that simply result from baseline differences in face shape.

In the light of these results, we speculate that it is necessary to study the expressions of emotions in each of the four quadrants resulting from the horizontal and vertical axes by distinguishing between spontaneous and posed displays, and investigating the function for which they are expressed and the type of anatomical pathway (i.e., Voluntary vs. Involuntary) underlying them, before we can draw a firm conclusion. Recent research points, indeed, at the existence of multiple interrelated networks, each associated with the processing of a specific component of emotions (i.e., generation, perception, regulation), which do not necessarily share the same lateralization patterns [50]. A recent meta-analysis revealed, in fact, that the perception, experience, and expression of emotion are each subserved by a distinct network [51]. Hence, the lateralization of emotion is a multi-layered phenomenon and, as such, should be considered.

### 6.2. Posed vs. Spontaneous

Results from two experiments demonstrated and confirmed that facial movements provide relevant and consistent details to characterize and distinguish between spontaneous and posed expressions. In line with our predictions, the results revealed that the speed and amplitude of the mouth as it widens into a smile are greater in posed than genuine happiness. In particular, a posed smile is characterized by an increase in the smile amplitude, speed, and deceleration, as indicated by the cheilion pair of markers. As concerns the upper part of the face, the results showed a similar increase in the Maximum Distance of the eyebrows when the participants performed a posed smile compared to when they smiled spontaneously. These findings confirm and extend previous literature [35,37,52,53] by showing that performing a fake smile entails a speeded choreography of amplified movements both in the lower and upper parts of the face.

The main limitation of our experiment is that it was not possible to extract the reaction times of the individual side markers (e.g., the right vs the left cheilion) because the videoclips triggered several smiles in sequence, and it was not easy to define the end of one smile and the beginning of the next. This is a widespread problem with ecological paradigms. We therefore focused on the speed at which the pair of markers on the mouth moved away: we defined the start of the movement as the event in which the speed crossed a 0.2 mm/s threshold and remained above it for longer than 100 ms. This parameter guaranteed us replicability and rigor, as it was not influenced by possible random sub-movements of a single marker, and it accounted for any head movement. However, this procedure did not allow us to define on which side of the face posed and spontaneous smiles began. Further studies are needed to clarify this aspect. Another limitation is that dynamic stimuli (video clips) could not be adopted for the Posed condition as well, because, otherwise, the voluntarily generated expressions would have overlapped and mixed with the authentic ones, not giving us the ability to discern one from the other. In the past, the use of static, posed, and archetypical stimuli, such as the one we adopted here for the Posed condition, has provided high scientific control and repeatability, but at the cost of ecological validity [54]. Further studies are therefore needed to adopt ecologically valid paradigms: in everyday life, posed expressions are produced to be perceived by another person (e.g., when mothers exaggerate their facial movements to be recognized by their infant children). The next step will be, therefore, to adopt real contexts to induce posed emotional displays.

This approach raises two questions: What if posed expressions that often occur in everyday life are nonetheless genuine? At what point does the benefit of using ecologically valid paradigms balance out the increase in inter-individual variability? Emotion science is now facing a classic trade-off. We believe, however, that if the science of emotions were to remain still anchored in prototypical displays and static induction methods, it would not rise to the level of understanding the processes that evolved in response to real social contexts during the phylogenetic development of the human species. Having a comprehensive taxonomy of real emotion expression will help to formulate new theories with a greater degree of complexity (for a review, see [15]).

### 6.3. Emotional Induction vs. Motor Contagion

Our data on the Likert scale indicate that both Emotional Induction and Motor Contagion were effective in activating a felt emotion of happiness. Moreover, videos adopted for the Motor Contagion were rated as more intense than those for the Emotional Induction. However, the comparison analysis on the two experiments showed no kinematic differences on spontaneous expressions depending on the method. Spontaneous expressions seem, therefore, to be conveyed by an automatic pathway, which is difficult to modify. This result is in line with the notion that a genuine emotion originates from subcortical brain areas that provide excitatory stimuli to the facial nerve nucleus via extrapyramidal motor tracts (i.e., the Involuntary Pathway). Future studies are needed to apply this methodology to other emotions in order to accurately investigate the full range of subtle differences in facial expressions and the role played by the Involuntary Pathway in emotion expression.

### 6.4. Clinical applications

The possibility of discriminating the spontaneous vs. posed expressions of emotions by means of sophisticated analysis of facial movements has potential for future clinical applications. One example is the application of this technique to patients suffering from Parkinson’s disease (PD), who are characterized by a deficit in the expression of genuine emotions (i.e., amimia), but who are still able to intentionally produce emotional expressions [55], thus manifesting the automatic–voluntary dissociation that underlies the distinction between the Voluntary and Involuntary Pathways. From a neuroanatomical point of view, patients with PD present defective functioning of the basal ganglia [56]. The connections between the basal ganglia and the cerebral cortex form extrapyramidal circuits, which are divided into five parallel networks connected to: the frontal motor and oculomotor cortex, the prefrontal cortex dorsolateral, the anterior cingulate cortex, and the orbitofrontal cortex [57]. Connections with the orbitofrontal and cingulate cortex constitute the limbic circuits, which also involve subcortical structures, such as the amygdala and hippocampus [58]. This rich neural network between the basal ganglia and the structures of the limbic system establishes a link between the perception of emotions, their motor production through facial expressions, and final recognition [59]. An in-depth investigation of the ability of PD patients to produce spontaneous and posed expressions, using an advanced and validated protocol for emotion induction and a sophisticated technique for data acquisition and analysis, could, therefore, be applied to investigate the emotion expression deficits in these patients. Furthermore, future research could investigate the correlation between the deficit in emotion production and functional/structural alterations of the brain in PD patients, in order to identify a behavioral biomarker that can estimate the severity of the disease.

In applicative terms, once the Clepsydra model has been tested on all basic emotions and validated on a large population including people of different ethnicities, it will be possible to use it through automatic facial recognition applications that are already present in the latest smartphones. The model will then allow it to be applied on a large scale and also reveal subtle changes in the expression of mixed emotions (e.g., happiness and surprise).

## 7. Conclusions

Despite the importance of emotions in human functioning, scientists have been unable to reach a consensus on the debated issue of lateralization of emotions. Our research, conducted with a 3-D high-definition optoelectronic system in conjunction with FACS, showed that posed smiles were more amplified than spontaneous smiles, with maximum acceleration occurring first in the left hemiface, followed by an earlier deceleration peak in the right corner of the mouth. This result would be in line with a recent hypothesis that the temporal dynamics of movement distinguish a posed smile from a spontaneous one: according to Ross and colleagues, posed expressions would, in fact, begin on the right side of the face [25]. The overall integration of this knowledge with our new data seems to suggest that posed smiles might begin in the right facial area, then rapidly expand into the contralateral hemiface before completing movement in the right side of the face. This loop seems to indicate dual hemispheric innervation along the vertical axis, and we can infer that both hemispheres exert motor control to produce an apparently unified expression [10].

Our findings may be the key to resolving the apparent conflict between various theories that have attempted to discriminate true and false expressions of happiness, and it will aid in clarifying the hemispheric bases of emotion expression. We believe, indeed, that investigating the dynamic pattern of facial expressions of emotions, which can be controlled consciously only in part, would provide a useful operational test for comparing the different predictions of various lateralization hypotheses, thus allowing this long-standing conundrum to be solved.

## Figures and Tables

**Figure 1 biology-12-01160-f001:**
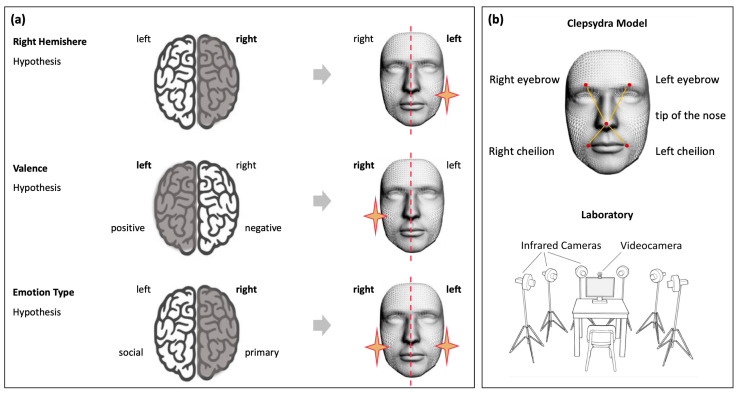
Schematic representation of the three main hypotheses of emotional processing and related patterns of facial lateralization (panel (**a**)). The Clepsydra Model (panel (**b**), top image) was adopted to test these hypotheses. Five markers were applied to the left and right eyebrows, left and right cheilions, and the tip of the nose. The experimental set up was equipped with six infrared cameras placed in a semicircle (bottom figure).

**Figure 2 biology-12-01160-f002:**
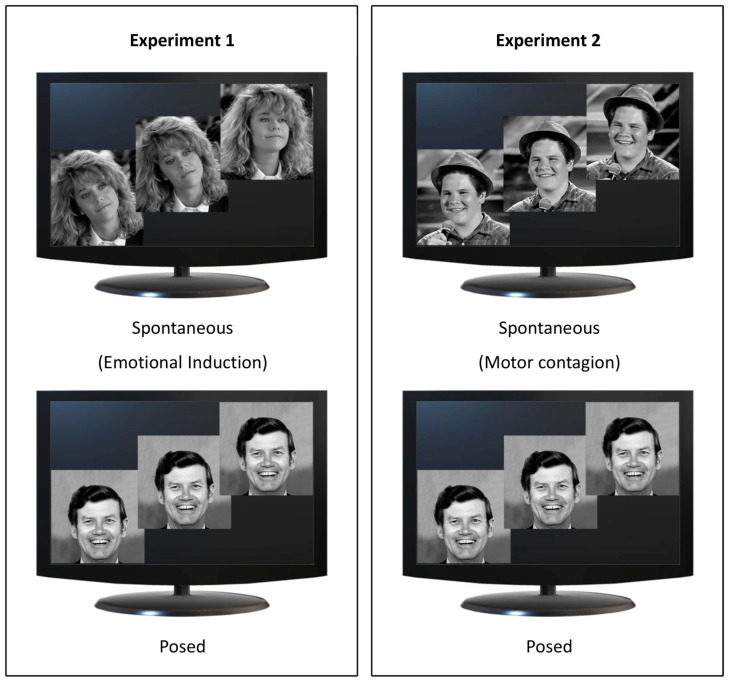
Experimental design. Spontaneous conditions for Experiments 1 and 2 are represented in the top panels and Posed conditions in the bottom panels. In Experiment 1 (**left** panel), participants viewed video extracts from comedy for Emotional induction (Spontaneous condition, upper image; source: YouTube) and a static image of happiness (Posed condition, lower image; source: Pictures of Facial Affect [39]). In Experiment 2 (**right** panel), participants viewed videos showing happy faces inducing Motor Contagion (Spontaneous condition, upper image; source: YouTube) and the same static image of happiness adopted in Experiment 1 (Posed condition, lower image; Pictures of Facial Affect [39]).

**Figure 3 biology-12-01160-f003:**
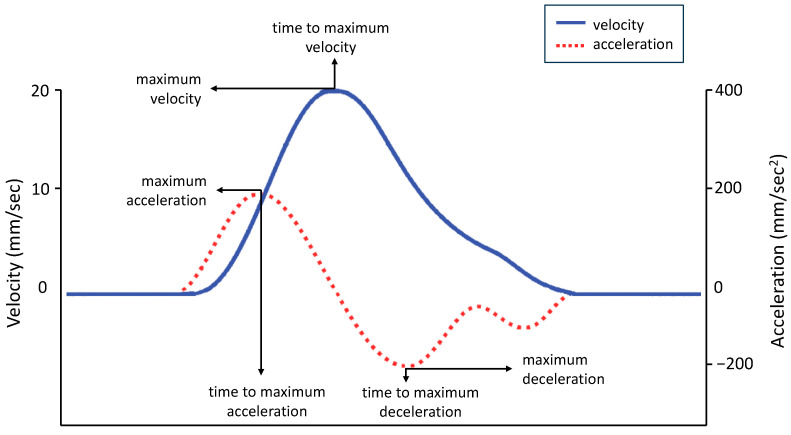
Graphical representation of experimental variables. The graphs of velocity (solid blue line) and acceleration (dashed red line) show the average amplitude and time sequence of the different peaks. Velocity events performed by the body classically occur in a predefined order: acceleration to peak velocity followed by deceleration. The same pattern is shown by the hand during a reaching task (for a comparison, see [43]).

**Figure 4 biology-12-01160-f004:**
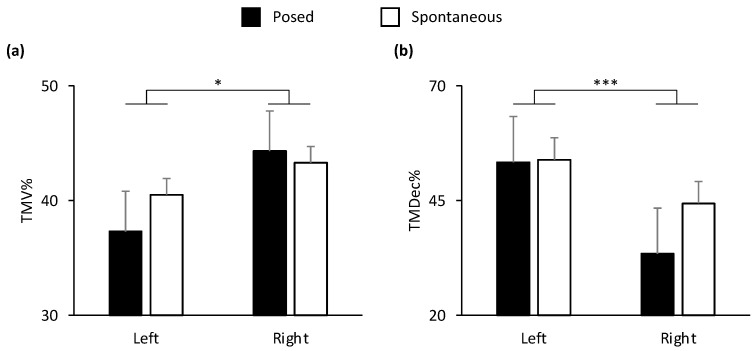
Graphical representation of temporal components of movement in the lower part (i.e., cheilion markers, CH) of the face during posed and spontaneous expressions of happiness. A significant main effect of side of the face was found for: (**a**) Time to Maximum Velocity (TMV%) and (**b**) Time to Maximum Deceleration (TMDec%) in the lower part of the face. TMV% was earlier in the left side of the face, and TMDec% was earlier in the right side of the face. Error bars represent standard error. Asterisks indicate statistically significant comparisons (* = *p* < 0.05; *** = *p* < 0.001).

**Figure 5 biology-12-01160-f005:**
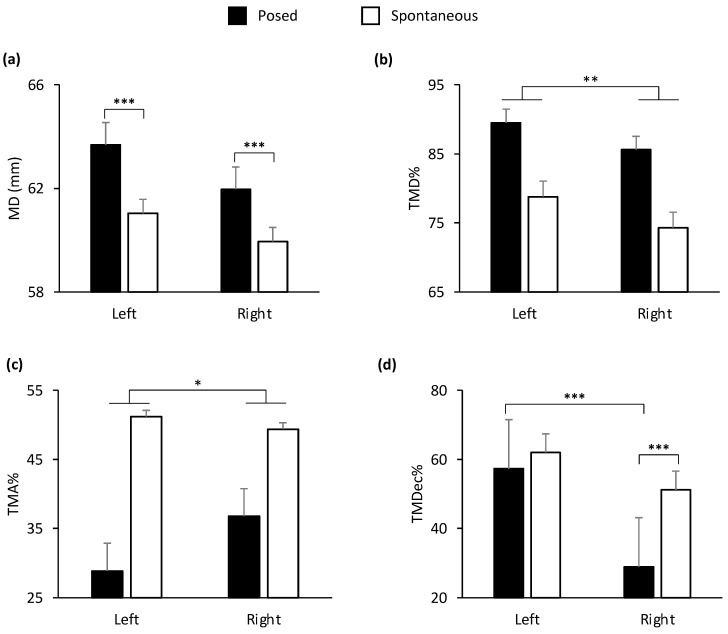
Graphical representation of spatial and temporal components of movement in the lower part (i.e., cheilion markers, CH) of the face during posed and spontaneous expressions of happiness. A main effect of side of the face (Left vs. Right) was shown for: (**b**) Time to Maximum Distance (TMD%) and (**c**) Time to Maximum Acceleration (TMA%). TMD% was earlier in the right side of the face and TMA% was earlier in the left side of the face. A statistically significant interaction condition by side of the face was found for: (**a**) Maximum Distance (MD) and (**d**) Time to Maximum Deceleration (TMDec%). MD was wider during posed compared to a spontaneous smile both in the left and right sides of the face. TMDec% of the right cheilion during posed smiles was the earliest, compared to both the left cheilion and the same marker during spontaneous smiles. Error bars represent standard error. Asterisks indicate statistically significant comparisons (* = *p* < 0.05; ** = *p* < 0.01; *** = *p* < 0.001).

**Figure 6 biology-12-01160-f006:**
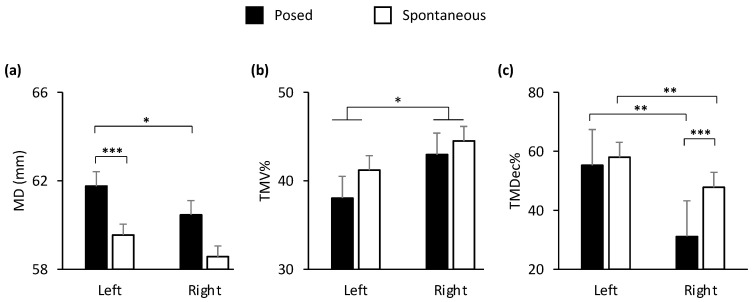
Graphical representation of spatial and of temporal components of movement in the lower part (i.e., cheilion markers, CH) of the face during posed and spontaneous expressions of happiness. A main effect of side of the face (Left vs. Right) was shown for: (**b**) Time to Maximum Velocity (TMV%). TMV% was reached earlier in the left than the right cheilion (**b**). A statistically significant interaction condition by side of the face was found for: (**a**) Maximum Distance (MD) and (**c**) Time to Maximum Deceleration (TMDec%). MD of the left cheilion during posed smiles was the widest, compared to both the right cheilion and the same marker during spontaneous smiles (**a**). TMDec% during spontaneous smiles was earlier in the right than in the left side of the face (**c**). TMDec% of the right cheilion during posed smiles was the earliest, compared to both the left cheilion and the same marker during spontaneous smiles (**c**). Error bars represent standard error. Asterisks indicate statistically significant comparisons (* = *p* < 0.05; ** = *p* < 0.01; *** = *p* < 0.001).

**Table 1 biology-12-01160-t001:** Results of Repeated-measures ANOVA for Experiment 1. Only parameters with at least one significant result were reported. Results on the main effect of condition are graphically represented in Appendix A enclosed in Appendix A. Two main effects of side of the face were found in the lower part of the face (see Figure 4). No interaction was statistically significant.

Kinematic Parameters	Main Effect Condition	Main Effect Side of the Face	Interaction Condition by Side of the Face
**Cheilions (CH)**			
**MD**	**F_(1,16)_ = 21.440, p < 0.001, ** **VS-MPR = 161.690, η^2^_p_ = 0.573**	F_(1,16)_ = 3.007, p = 0.102, VS-MPR = 1.579, η^2^_p_ = 0.158	F_(1,16)_ = 0.014, p = 0.908, VS-MPR = 1.000, η^2^_p_ < 0.001
**DD**	**F_(1,16)_ = 8.221, p = 0.011, ** **VS-MPR = 7.325, η^2^_p_ = 0.339**	F_(1,16)_ = 1.882, p = 0.189, VS-MPR = 1.168, η^2^_p_ = 0.105	F_(1,16)_ = 1.23, p = 0.305, VS-MPR = 1.016, η^2^_p_ = 0.066
**MV**	**F_(1,16)_ = 10.595, p = 0.005, ** **VS-MPR = 13.958, η^2^_p_ = 0.398**	F_(1,16)_ = 0.636, p = 0.437, VS-MPR = 1.000, η^2^_p_ = 0.038	F_(1,16)_ = 0.539, p = 0.473, VS-MPR = 1.000, η^2^_p_ = 0.033
**MA**	**F_(1,13)_ = 8.523, p = 0.012, ** **VS-MPR = 6.952, η^2^_p_ = 0.396**	F_(1,13)_ = 0.365, p = 0.556, VS-MPR = 1.000, η^2^_p_ = 0.027	F_(1,13)_ = 0.029, p = 0.868, VS-MPR = 1.000, η^2^_p_ = 0.002
**MDec**	**F_(1,13)_ = 6.491, p = 0.024, ** **VS-MPR = 4.073, η^2^_p_ = 0.333**	F_(1,13)_ = 0.766, p = 0.397, VS-MPR = 1.000, η^2^_p_ = 0.056	F_(1,13)_ = 0.192, p = 0.668, VS-MPR = 1.000, η^2^_p_ = 0.015
**TMD%**	**F_(1,16)_ = 5.670, p = 0.030, ** **VS-MPR = 3.495, η^2^_p_ = 0.262**	F_(1,16)_ = 0.026, p = 0.873, VS-MPR = 1.000, η^2^_p_ = 0.002	F_(1,16)_ = 1.142, p = 0.301, VS-MPR = 1.018, η^2^_p_ = 0.067
**TMV%**	F_(1,16)_ = 0.120, p = 0.733, VS-MPR = 1.000, η^2^_p_ = 0.007	**F_(1,16)_ = 4.616, p = 0.047, ** **VS-MPR = 2.548, η^2^_p_ = 0.224**	F_(1,16)_ = 0.530, p = 0.477, VS-MPR = 1.000, η^2^_p_ = 0.032
**TMA%**	**F_(1,14)_ = 5.670, p = 0.030, ** **VS-MPR = 3.495, η^2^_p_ = 0.262**	F_(1,14)_ = 0.709, p = 0.414, VS-MPR = 1.000, η^2^_p_ = 0.048	F_(1,14)_ = 0.562, p = 0.466, VS-MPR = 1.000, η^2^_p_ = 0.039
**TMDec%**	F_(1,14)_ = 1.168, p = 0.298, VS-MPR = 1.020, η^2^_p_ = 0.077	**F_(1,14)_ = 24.37, p < 0.001, ** **VS-MPR = 188.689, η^2^_p_ = 0.632**	F_(1,14)_ = 2.795, p = 0.117, VS-MPR = 1.467, η^2^_p_ = 0.166
**Eyebrows (EB)**			
**MD**	**F_(1,16)_ = 12.298, p = 0.003, ** **VS-MPR = 21.580, η^2^_p_ = 0.435**	F_(1,16)_ = 0.518, p = 0.482, VS-MPR = 1.000, η^2^_p_ = 0.031	F_(1,16)_ = 1.411, p = 0.252, VS-MPR = 1.059, η^2^_p_ = 0.081
**TMV%**	**F_(1,14)_ = 10.083, p = 0.007, ** **VS-MPR = 10.912, η^2^_p_ = 0.419**	F_(1,14)_ = 0.287, p = 0.601, VS-MPR = 1.000, η^2^_p_ = 0.020	F_(1,14)_ = 0.413, p = 0.531, VS-MPR = 1.000, η^2^_p_ = 0.029

Statistically significant data are shown in bold (p < 0.05).

**Table 2 biology-12-01160-t002:** Results of Repeated-measures ANOVA for Experiment 2. Only parameters with at least one significant result were reported. Results on the main effect of condition are graphically represented in Appendix A enclosed in Appendix A. Four main effects of side of the face and two interactions were found in the lower part of the face (see Figure 5).

Kinematic Parameters	Main Effect Condition	Main Effect Side of the Face	Interaction Condition by Side of the Face
**Cheilions (CH)**			
**MD**	**F_(1,19)_ = 29.400, p < 0.001, ** **VS-MPR = 1135.133, η^2^_p_ = 0.607**	**F_(1,19)_ = 5.681, p = 0.028, ** **VS-MPR = 3.700, η^2^_p_ = 0.230**	**F_(1,19)_ =6.452, p = 0.020, ** **VS-MPR = 4.706, η^2^_p_ = 0.253**
**DD**	**F_(1,19)_ = 21.393, p < 0.001, ** **VS-MPR = 231.784, η^2^_p_ = 0.530**	F_(1,19)_ = 0.187, p = 0.670, VS-MPR = 1.000, η^2^_p_ = 0.010	F_(1,19)_ = 0.080, p = 0.780, VS-MPR = 1.000, η^2^_p_ = 0.004
**MV**	**F_(1,19)_ = 29.728, p < 0.001, ** **VS-MPR = 1205.041, η^2^_p_ = 0.610**	F_(1,19)_ = 3.451, p = 0.079, VS-MPR = 1.837, η^2^_p_ = 0.154	F_(1,19)_ = 0.165, p = 0.689, VS-MPR = 1.000, η^2^_p_ = 0.009
**MA**	**F_(1,19)_ = 17.149, p < 0.001, ** **VS-MPR = 88.406, η^2^_p_ = 0.474**	F_(1,19)_ = 0.102, p = 0.753, VS-MPR = 1.000, η^2^_p_ = 0.005	F_(1,19)_ = 0.273, p = 0.608, VS-MPR = 1.000, η^2^_p_ = 0.014
**MDec**	**F_(1,19)_ = 18.450, p < 0.001, ** **VS-MPR = 120.051, η^2^_p_ = 0.493**	F_(1,19)_ = 0.473, p = 0.500, VS-MPR = 1.000, η^2^_p_ = 0.024	F_(1,19)_ = 0.895, p = 0.356, VS-MPR = 1.001, η^2^_p_ = 0.045
**TMD%**	**F_(1,19)_ = 26.586, p < 0.001, ** **VS-MPR = 669.279, η^2^_p_ = 0.583**	**F_(1,19)_ = 9.818, p = 0.005, ** **VS-MPR = 12.910, η^2^_p_ = 0.341**	F_(1,19)_ = 0.036, p = 0.851, VS-MPR = 1.000, η^2^_p_ = 0.002
**TMA%**	**F_(1,19)_ = 17.956, p < 0.001, ** **VS-MPR = 106.987, η^2^_p_ = 0.486**	**F_(1,19)_ = 5.300, p = 0.033, ** **VS-MPR = 3.282, η^2^_p_ = 0.218**	F_(1,19)_ = 4.089, p = 0.057, VS-MPR = 2.241, η^2^_p_ = 0.177
**TMDec%**	**F_(1,19)_ = 10.120, p = 0.005, ** **VS-MPR = 14.076, η^2^_p_ = 0.348**	**F_(1,19)_ = 46.466, p < 0.001, ** **VS-MPR = 16685.144, η^2^_p_ = 0.710**	**F_(1,19)_ = 9.707, p = 0.006, ** **VS-MPR = 12.502, η^2^_p_ = 0.338**

Statistically significant data are shown in bold (*p* < 0.05).

**Table 3 biology-12-01160-t003:** Results of Mixed ANOVA (comparison analysis). Only parameters with at least one significant result were reported. Results on the main effect of condition are graphically represented in Appendix A enclosed in Appendix A. Three main effects of side of the face and two interactions were found in the lower part of the face (see Figure 6).

Kinematic Parameters	Main Effect Condition	Main Effect Side of the Face	2-Way Interaction between Condition and Side of the Face
**Cheilions (CH)**			
**MD**	**F_(1,35)_ = 49.138, p < 0.001, ** **VS-MPR = 579,497.156, η^2^_p_ = 0.584**	**F_(1,35)_ = 8.314, p = 0.007, ** **VS-MPR = 10.987, η^2^_p_ = 0.192**	**F_(1,35)_ = 4.106, p = 0.05, ** **VS-MPR = 2.443, η^2^_p_ = 0.105**
**DD**	**F_(1,35)_ = 27.775, p < 0.001, ** **VS-MPR = 4382.16, η^2^_p_ = 0.442**	F_(1,35)_ = 1.380, p = 0.248, VS-MPR = 1.064, η^2^_p_ = 0.038	F_(1,35)_ = 0.487, p = 0.490, VS-MPR = 1.000, η^2^_p_ = 0.014
**MV**	**F_(1,35)_ = 36.953, p < 0.001, ** **VS-MPR = 42,283.314, η^2^_p_ = 0.514**	F_(1,35)_ = 3.246, p = 0.080, VS-MPR = 1.817, η^2^_p_ = 0.085	F_(1,35)_ = 0.167, p = 0.685, VS-MPR = 1.000, η^2^_p_ = 0.005
**MA**	**F_(1,32)_ = 23.699, p < 0.001,** **VS-MPR = 1208.896, η^2^_p_ = 0.425**	F_(1,32)_ = 0.498, p = 0.485, VS-MPR = 1.000, η^2^_p_ = 0.015	F_(1,32)_ = 0.031, p = 0.861, VS-MPR = 1.000, η^2^_p_ < 0.001
**MDec**	**F_(1,32)_ = 22.148, p < 0.001, ** **VS-MPR = 791.644, η^2^_p_ = 0.409**	F_(1,32)_ =0.038, p = 0.847, VS-MPR = 1.000, η^2^_p_ = 0.001	F_(1,32)_ = 0.898, p = 0.350, VS-MPR = 1.001, η^2^_p_ = 0.027
**TMD%**	**F_(1,35)_ = 27.941, p** ** < 0.001, VS-MPR = 4576.896, η^2^_p_ = 0.444**	F_(1,35)_ =3.258, p = 0.080, VS-MPR = 1.825, η^2^_p_ = 0.085	F_(1,35)_ = 1.128, p = 0.296, VS-MPR = 1.021, η^2^_p_ = 0.031
**TMV%**	F_(1,35)_ =1.136, p = 0.294, VS-MPR = 1.022, η^2^_p_ = 0.031	**F_(1,35)_ = 6.551, p = 0.015, ** **VS-MPR = 5.851, η^2^_p_ = 0.158**	F_(1,35)_ =0.200, p = 0.657, VS-MPR = 1.000, η^2^_p_ = 0.006
**TMA%**	**F_(1,33)_ = 20.198, p < 0.001, ** **VS-MPR = 481.057, η^2^_p_ = 0.380**	F_(1,33)_ = 3.389, p = 0.075, VS-MPR = 1.900, η^2^_p_ = 0.093	F_(1,33)_ = 3.683, p = 0.064, VS-MPR = 2.098, η^2^_p_ = 0.100
**TMDec%**	**F_(1,33)_ = 8.160, p = 0.007, ** **VS-MPR = 10.181, η^2^_p_ = 0.198**	**F_(1,33)_ = 66.159, p < 0.001, ** **VS-MPR > 100,000, η^2^_p_ = 0.667**	**F_(1,33)_ = 10.947, p = 0.002, ** **VS-MPR = 26.596, η^2^_p_ = 0.249**
**Eyebrows (EB)**			
**MD**	**F_(1,35)_ = 6.535, p = 0.015, ** **VS-MPR = 5.818, η^2^_p_ = 0.157**	F_(1,35)_ < 0.01, p = 0.986, VS-MPR = 1.000, η^2^_p_ < 0.001	F_(1,35)_ = 2.596, p = 0.116, VS-MPR = 1.471, η^2^_p_ = 0.069

Statistically significant data are shown in bold (p < 0.05).

## Data Availability

The dataset has been uploaded in the Appendix A section.

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
