# Peer review of "The Spatiotemporal Dynamics of Facial Movements Reveals the Left Side of a Posed Smile"

_biology, 2023, doi:10.3390/biology12091160_

Round 1

Reviewer 1 Report (Previous Reviewer 2)

All of my concerns have been addressed in the revised version.

Author Response

Thanks

Reviewer 2 Report (New Reviewer)

This paper proposes certain spatial and speed key kinematic patterns as the reliable parameters for the distinguishing spontaneous expressions from posed facial movements.  Paper is well organized, comprehensible and can be easily followed. Experiment procedure and hardware are meticulously described.

However, there are some questions and remarks concerning the key points of the paper:

Paragraph 2.5.1.

1. Maximum Distance is maximum distance between two markers, that is clear. But, what is the maximum velocity? Is it relative velocity of one marker with respect to the other one?

2.  How do you calculate velocities and, especially, accelerations? In reference (40) velocity is briefly mentioned, there is no mention of the accelerations. Same for normalized values, it would be good to give the equations.

Paragraph 2.6.

3. Why do you adopt sample size of 20 individuals, it looks small for such an experiment? Reference (43) is an old one – 1996. year, reference (35) is more recent one and author use sample of 400 individuals.

Paragraphs 3.3., 4.3. and 5

4. Figures and Tables must be better explained. All marks must be defined!

Paragraph 6.

5. Due to the poor presentation of the results it is hard to understand and accept the  claims stated in the discussion.

6. You mention acceleration and deceleration peaks, double peak etc., it would be good to present time history of the acceleration/deceleration.

7. Are there more application aspects?

Round 2

Reviewer 2 Report (New Reviewer)

Authors have answered the questions and explained the ambiguities. Additional material is added to the paper  which improves and clarifies its essence.

This manuscript is a resubmission of an earlier submission. The following is a list of the peer review reports and author responses from that submission.

Round 1

Reviewer 1 Report

This MS discusses the topic of facial expressions in humans and the underlying anatomical pathways involved in modulating these expressions. It also highlights the distinction between voluntary (VP) and involuntary pathways (IP) in producing posed and spontaneous facial expressions, as well as their potential differential effects on the left and right sides of the face. Two experiments are presented to investigate the unfolding of spontaneous and posed facial expressions of happiness along the facial vertical axis (left and right sides), using different induction methods. These experiments utilized a high-definition 3D optoelectronic system to capture facial movements. The findings revealed distinguishable patterns between spontaneous and posed expressions in both experiments. Furthermore, it was observed that the VP activation resulted in a lateralization effect. Specifically, compared to a genuinely felt smile, a posed smile involved an initial acceleration of the left corner of the mouth, followed by an early deceleration of the right corner in the second phase of the movement, occurring after the velocity peak.

Overall, I found the topic of the manuscript interesting, and the methods employed sound enough to test hypotheses. At the same time, I was unclear regarding the original contribution of this study with respect to what has been done in the literature—at the theoretical and empirical level—and the extent to which the authors found results aligned to their initial hypotheses.

I recommend to the authors to better ground their hypotheses into previous literature, specifying what the original contribution of this work is. Furthermore, the authors could make clearer the extent to which their hypotheses were confirmed in the two studies.

Another important aspect that requires consideration is the power analysis and the choice to employ ‘only’ 20 participants per experiment. I would like to see a more thorough explanation of how the sample size was estimated, based in particular on the type of analyses conducted (two-ways within-subjects ANOVAs and two-way ANOVAs between-subjects). This is an important point as the significant results out of the multiple analyses might in fact be the result of chance.

Below I report more detailed comments for specific sections of the MS.

1.      - The paragraph from lines 108 to 117 mixes up the objective of the study with the operationalization of the variables, which is confusing. I encourage the authors to first explain what the manipulation was for exactly, and then link it directly to how it was operationalized in each experiment.

2.       - Also, it would be good to explain what the reason was for having two experiments and what were the hypothesis for each of them. It seems to me that in the end it was not two experiments, but one experiment with 1 between-subjects variable and 1 within-subjects condition.

3.       -  2.6 Data Analysis should rather be data acquisition.

4.       -  2.7 Statistical Analysis should rather be Statistical (or analytical) approach.

5.       -  I think the GPOWER calculation should have been done separately for the different analyses, as in the first case it was a repeated measure ANOVA, but in the second case a mixed ANOVA. It would be good if the authors could attach to the revision the output of GPOWER to support the choice of 20 participants per experiment.

6.      -  Lines 296-297: “The stimuli were 296 judged to have above-average intensity (6; SD = 1,711) on a 9-point Likert scale.” How did the authors know that the intensity was above average? What was the control/baseline condition to compare with?

7.      -  In experiment 2, despite being stimuli more intense, no interaction was significant and especially the results for the upper side of the face were ns. How do the authors explain this?

8.       - The two experimental conditions in each experiment were chosen to activate the two pathways: Voluntary and Involuntary. However, results do not speak about the two of them, but just one. Also, significant interactions were hypothesized, but only few significant results were found. Having a discussion around this point would be helpful for the reader to better understand  

9.       -  I think the authors could be more transparent to summarize findings that support hypotheses vs. significant findings not directly addressing hypotheses, but that may nevertheless be interesting to report.

English level was good in my opinion. Poroofreading necessary.

Reviewer 2 Report

After reading this article, I acknowledge your work. I think your method has potential application value. But there are still many shortcomings, and I briefly list some of the shortcomings of your research below.

1. Specific mathematical explanations should be provided for the relevant parameters. (Section 2.6.2)

2. Suggest presenting the results of variance measurement in the form of a chart. (Section 3.3)

3. Specific explanations should be provided for the results in Figure 2 and Figure 3.

4. Is the conclusion about upper part of the face too absolute? (Section4.3.1)

5. Please provide appropriate proof of this conclusion. (Section4.3.2)

6. Comparative analysis suggests presenting in table form. (Section 5.1)

7. The method section did not provide specific formulas for calculating the results, and the experimental data lacked rigorous theoretical support. (Section 2)

8. There is too much text in the experimental and comparative sections. It is recommended to provide a detailed explanation of the results in the form of a chart.

9. The experimental part is not detailed enough, and the conclusions drawn may not be sufficient to support your theory. (Sections 6.1, 6.2)

10. The conclusion section suggests summarizing the shortcomings of one's own research and providing future recommendations (Conclusions)

Minor editing of English language required.

Reviewer 3 Report

Straulino and colleagues showed that spontaneous facial expressions were distinguished from posed facial movements and that the latter produced a lateralization effect. The methods are well described, and the two behavioral tasks are conducted in a rightful manner. However, it is not entirely clear to me the novelty of the study and the general interest in its take-home message. The authors could expand the research to other emotional states to extend the knowledge in the field.